# Protocol for a multicentre, prospective observational cohort study in Japan: association among hospital-acquired disability, regular exercise and long-term care dependency in older patients after cardiac surgery

Masakazu Saitoh [1], Tetsuya Takahashi,[1] Tomoyuki Morisawa,[1] Akihiro Sakuyama,[1] Hidetaka Watanabe,[2] Koji Sakurada,[3] Yusuke Hanafusa,[4] Masayuki Tahara,[5] Kentaro Iwata,[6] Yusuke Ochi,[7] Go Takamura,[8] Akira Minei[9]

For numbered affiliations see end of article.

**Correspondence to**
Professor Tetsuya Takahashi; te-takahashi@juntendo.ac.jp

## ABSTRACT

**Introduction** Cardiac surgery for older patients, postoperative functional decline and the need for long-term care have received increasing attention as essential outcomes in recent years. Therefore, prevention of functional decline and long-term care dependency after cardiac surgery are important; however, our current understanding of postoperative functional trajectory and effects of postoperative regular exercise on long-term functional decline and long-term care dependency is limited. Therefore, we will conduct a multicentre, prospective cohort study to (1) examine the effect of hospital-acquired disability on long-term functional decline and long-term care dependency and (2) investigate the favourable effect of postoperative regular exercise on long-term functional decline and long-term care dependency in older patients after cardiac surgery.

**Methods and analysis** We designed a prospective, multicentre cohort study to enrol older patients aged≥65 years undergoing elective coronary artery bypass graft or valve surgery. We will conduct medical record reviews to collect data on patient demographics, comorbidities, operative details, progression of in-hospital postoperative cardiac rehabilitation and functional trajectory from a few days before cardiac surgery to the day before hospital discharge. They will be followed up for 2 years to obtain information on their health status including functional status, regular exercise and clinical events by mail. Primary endpoints of this study are long-term functional decline and long-term care dependency after cardiac surgery. Secondary endpoints are readmission due to cardiac events or all-cause mortality.

**Ethics and dissemination** The study protocol was approved by the Ethics Committee of the Department of Physical Therapy, Faculty of Health Science, Juntendo University, and of each collaborating hospital. We obtained written informed consent from all study participants after the description of the study procedures. Publication of the study results is anticipated in 2025.

## Strength and limitations of this study

► As the first prospective, multicentre cohort study will provide important evidence on the trajectory of the functional status, long-term care dependency in older patients undergoing elective cardiac surgery.
► Risk factors for hospital-acquired disability in older patients undergoing elective cardiac surgery will be investigated.
► Prospective follow-up will help to identify new strategy associated with trajectory of the long-term care dependency and long-term health status.
► This study is not population based due to the recruitment scheme in collaborative hospital that focuses on research and education in the field of cardiac surgery rehabilitation, which limits its generalisability and underestimate the prevalence of functional decline.
► An important potential limitation is selection bias introduced by the inclusion of individuals who meet the eligibility criteria and are capable of functional assessments or answering the questionnaires on functional status.

## INTRODUCTION

Japan is now categorised as a super-aged society, with a population of individuals aged≥65 years being 35.89 million (28.4%),[1] which will reach 30% in 2025 and 40% in 2060.[2] The Japanese Association for Thoracic Surgery has conducted annual surveys of thoracic surgery throughout Japan. Cardiac surgery is the most common of thoracic surgery, more than 70 000 patients in Japan undergo cardiac surgery annually,[3] in particularly the absolute number of older patients undergoing cardiac surgery is increasing.[4] In

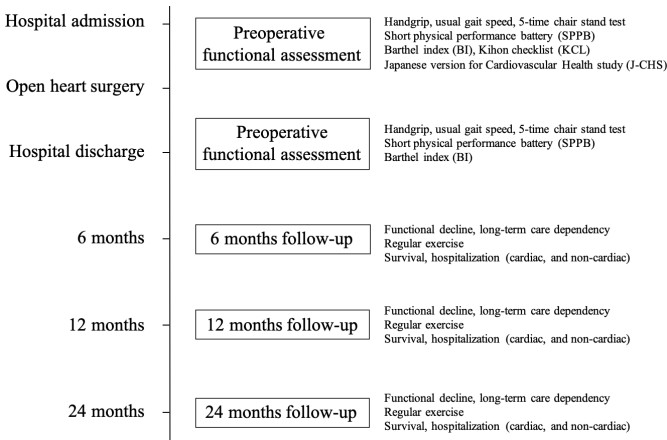

**Figure 1** Experimental protocol.

the consideration of cardiac surgery for older patients, functional decline and the need for long-term care have received increasing attention as essential outcomes in recent years.

Hospital-acquired disability (HAD) or functional decline, which refers to either a new or worsening disability or functional decline during hospitalisation not present before hospitalisation, develops in 25%–50% of hospitalised elderly patients.[5 6] In-hospital complications or comorbidities could lead to a longer hospitalisation stay after cardiac surgery.[7–9] Although the impact of HAD on long-term functional decline and long-term care dependency may be significant, it has not been investigated in detail yet.

Most studies have not prospectively measured functional trajectory from baseline to hospital discharge and long-term follow-up.[10 11] In addition, the need for long-term care, which is defined as a condition wherein older patients need care to perform activities of daily living (ADLs), increases with age and disease. The number of individuals who need long-term care in Japan has increased, leading to increasing social security expense for long-term care.[12] Thus, the Japanese government launched a policy with the aim to extend healthy life expectancy without the need for long-term care. In Japan, long-term care is required mainly for patients with dementia (18.7%), cerebrovascular disease (15.1%), age-related frailty (13.8%), falls and fractures (12.5%), joint disease (10.2%) and cardiac disease (4.7).[1] However, the trajectory for long-term care requirement after cardiac surgery remains unclear.

Recent meta-analyses showed that exercise-based cardiac rehabilitation (CR) improves exercise capacity in patients after cardiac surgery.[13 14] However, CR implementation rate for patients with acute myocardial infarction and heart failure is very low (21% and 7%, respectively).[15 16] In particular, there are few reports on CR implementation rate in older patients with frailty or functional disability after cardiac surgery.[17] However, several previous studies have demonstrated that CR or geriatric rehabilitation improves functional status and quality of life in older

patients with frailty or functional decline.[18 19] We hypothesised that functional trajectory after cardiac surgery and regular exercise after hospital discharge significantly impact long-term functional decline and long-term care dependency as well as morbidity and mortality; however, this hypothesis has not been tested yet.

In this context, we developed a multicentre prospective cohort study for elderly patients undergoing cardiac surgery. The primary objective of the present study is to prospectively investigate the association of functional trajectory after cardiac surgery and postdischarge regular exercise with clinical outcomes. The primary outcomes of the present study are long-term functional decline and long-term care dependency. Secondary outcomes are hospitalisation and all-cause mortality.

## METHODS AND ANALYSIS
### Study design and setting
The study is a multicentre, prospective cohort study that will be carried out between August 2021 and December 2024 in Japan enrolling older patients awaiting elective cardiac surgery at hospital admission; these patients will be followed up for 2 years after discharge. The overall study design is outlined in figure 1.

### Participant eligibility criteria
Potential participants are recruited from collaborating hospitals. To be eligible for study participation, patients must be aged≥65 years and scheduled to undergo elective cardiac surgery. Exclusion criteria are presented in figure 2.

### Collaborating hospitals
All collaborating hospitals provided a standardised in-hospital CR programme according to the Guidelines of Japanese Association of Cardiac Rehabilitation.[20] Functional assessments are performed by a trained physiotherapist according to the functional assessment manual.

### Measurements
The collaborating hospitals are encouraged to register the patients as consecutively as possible. Once informed consent is obtained, patients are assigned a study ID. For each patient, demographic data, aetiology of cardiac surgery, comorbidities, laboratory data, echocardiographic findings, medications and progression of postoperative in-hospital CR are collected. A mail survey on clinical outcomes and questionnaire on frailty, functional status and regular exercise (home-based non-supervised exercise, centre-based geriatric rehabilitation, centre-based CR) will be conducted at 6, 12 and 24 months after hospital discharge.

### Activity of daily living
The baseline Barthel Index (BI) was evaluated on the basis of interviews with patients or family members, depending on the patient's cognitive function, approximately 1 month before hospital admission, as in a previous

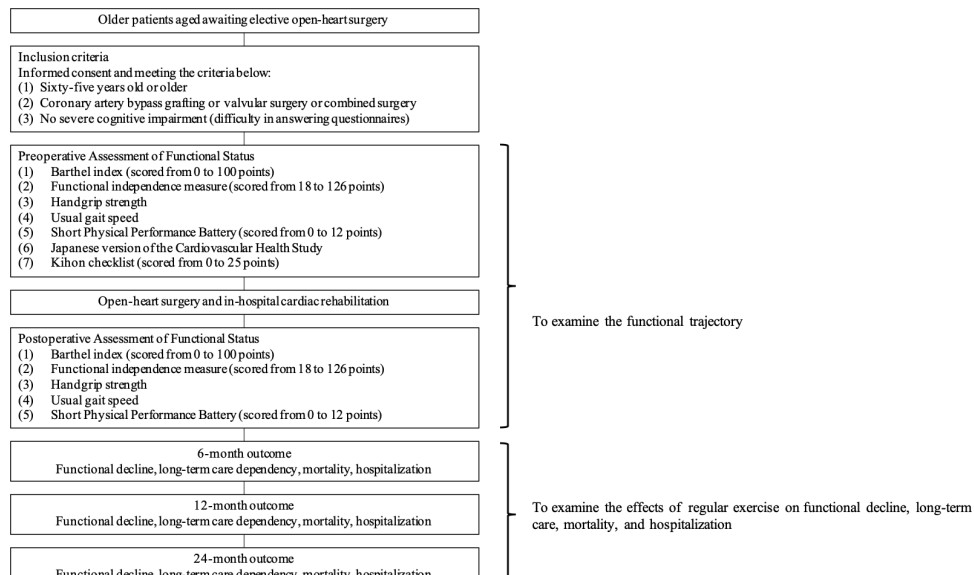

**Figure 2** Overall design of the study and main objectives.

study.[21] BI was evaluated by a trained physiotherapist at each hospital. BI is a simple 10-item instrument that measures functional independence in ADLs. The total score ranges from 0 to 100 points, with higher scores indicating greater independence in performing basic ADLs. HAD was defined as a decrease by at least five points on BI the day before discharge to home, nursing care facility or other hospitals compared with preadmission BI. Subanalyses were performed using information on no HAD, mild HAD as 5-point BI Score decrease and severe HAD as ≥10-point BI Score decrease.[21]

### Frailty assessment

The Kihon Checklist (KCL), a self-reported comprehensive health checklist, is a screening tool used to identify community-dwelling older adults who are vulnerable to frailty and at greater risk of becoming dependent.[22] The KCL consists of 25 items (yes/no) divided into seven categories: physical strength, nutrition, eating, socialisation, memory, mood and lifestyle; each item was scored from 0 to 1 point, and the sum of all indices can be 0–25; a total score of ≥8 points is classified as indicating frailty. In addition, seven specific domains of frailty—instrumental ADL, physical, nutrition, oral, socialisation, memory, and mood domains—were scored.

The Japanese version of the Cardiovascular Health Study (J-CHS) criteria were constructed by modifying and simplifying the original CHS criteria to be suited to Japanese older adults.[23] Moreover, J-CHS can predict a new occurrence of long-term care need and loss of independence. It consists of five items: (1) Shrinking: 'Have you unintentionally lost 2 kg or more in the past 6 months?' (2) Low activity: (a) 'Do you engage in moderate levels of physical exercise or sports aimed at health?' and (b) Do you engage in low levels of physical exercise aimed at health?' (3) Exhaustion: 'In the past 2 weeks, have you felt tired without a reason?' (4) Weakness: 'Defined as grip strength<28 kg in men and 18 kg in women'. (5) Slowness: 'Defined as gait speed of <1.0 m/s'. Each item was scored as 0 or 1 point and all points were added to calculate total score. A total score of ≥3 points was considered as indicating frailty.

### Short physical performance battery (SPPB)

The SPPB is a highly standardised geriatric physical functioning test that consists of tests for balance, gait, strength and endurance.[24] The standing balance test includes side-by-side, semitandem and tandem standing and patients are timed until they lose balance or 10 s elapses. The gait test assesses the time taken to walk 4 m and is performed at patient's usual pace. The 5-time chair-standing test, a pretest, is also performed, wherein patients are asked to fold their arms across their chest and stand up from the chair. If the pretest is successful, patients are asked to rise from the chair 5 times consecutively as quickly as possible. Each of the three subtests of SPPB is scored from 0 to 4 and summed for an SPPB Score of 0–12 points, with higher scores indicating better physical function.

### Hand grip strength

Isometric hand grip strength is measured using the Jamar hand dynamometer, which measures strength in kg with a precision of 0.1 kg. Patients are instructed to sit on a straight-backed chair with feet placed flat on the floor, shoulder adducted and neutrally rotated, elbow flexed at 90° and forearm and wrist in neutral position, as recommended by Roberts *et al*. Grip strength is measured twice per hand in each of the two handle positions.[25] Patients are instructed to apply maximum power for 3 s and work at maximum effort in every trial.

### In-hospital postoperative CR

The acute phase of the CR programme following cardiovascular surgery has been described elsewhere[9] and is

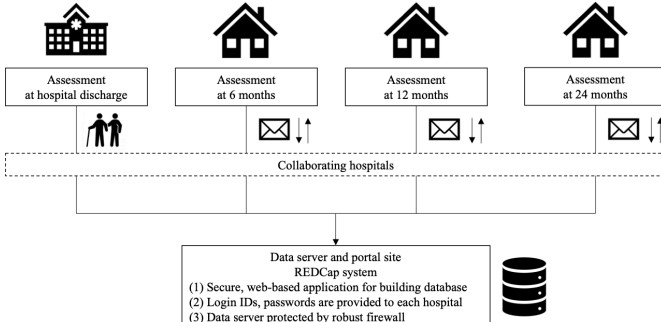

**Figure 3** Data registration system and database security.

based on the Japanese Circulation Society guidelines for rehabilitation of patients with cardiovascular disease.[20] We, therefore, define failure to achieve early ambulation as ≥3 postoperative days taken to initiate walking exercise according to the guidelines. Moreover, the proportion of patients who can complete a 100m corridor walk without assistance at a comfortable pace within 3 days of hospital admission is investigated; early ambulation is an important indicator of mobility or physical activity recovery during hospitalisation.[26]

### Outcome measures

The primary endpoints of this study are functional status trajectory and long-term care dependency. Functional status is measured using the KCL total score and specific seven-domain score. Secondary endpoints are readmissions because of cardiac events and all-cause mortality.

### Sample size calculation

Sample size is calculated by performing multivariate analysis, which examines the relationship among functional trajectory, regular exercise, physical activity and clinical outcomes. Eight independent variables, including functional trajectory, regular exercise and known prognostic factors of patients undergoing cardiac surgery, are selected for multivariate analysis. Assuming 10 outcomes per independent variable, 80 outcomes need to be observed. Based on our preliminary data, 8% incidence of a cardiac event in 2 years was estimated. With a 10% missing follow-up data, the necessary sample size was calculated to be 1100 patients. Because we also plan to perform a cross-sectional analysis using baseline data, a preliminary sample size calculation was done. Functional trajectory after cardiac surgery related clinical outcomes including morbidity, mortality, functional decline, and long-term care. We estimated that HAD incidence in the group with delayed in-hospital CR was 30%, compared with 20% in the group with normal progression of the guideline-based CR. A sample size of 1092 patients was estimated to be necessary based on the statistical power of 80% at a significance level of 5%. Thus, the estimated required number of patients in the cross-sectional analysis is sufficient for the prospective analysis. Therefore, we set the overall sample size as 1100 in the present study.

### Statistical analysis

Continuous variables will be presented through centrality measures (mean or median) and dispersion (SD or IQR), and categorical variables will be presented through frequencies and percentages. Statistical analysis will be performed to identify risk factors of functional trajectory after cardiac surgery. The resulting dataset is a complex time-series data, with each patient having three data points (preoperative, Postoperative day (POD) 7 and hospital discharge) for each physical performance measures. The trajectories can be labelled according to the observed clinical phenotype as below: complete recovery, partial recovery and ADL dependence in each patient with and without physical frailty. This data-driven classification allows the identification of predictors in patients belonging to a certain class of functional trajectory path. To identify predictors of functional trajectories, multinominal logistic regression models will be applied using preoperative demographics, comorbidities, perioperative and postoperative variables and baseline functional status or frailty. Multivariate Cox regression analysis will be used to separately model the risk of (1) 2-year functional decline, (2) 2-year long-term care dependency level, (3) 2-year all-cause mortality and (4) 2-year cardiovascular event.

Missing data on endpoints will not be replaced. A replacement method, multiple imputations through chained equations will be envisaged for other variables and other variables. A p value<0.05 will be considered statistically significant.

### Data availability

All patient data are stored in a secure web-based database (REDCap) with limited access and ID code to which data are directly transferred. Patients are assigned unique identification numbers, which are the only identifiers exported from REDCap during data analysis (figure 3). The principal investigator has access to all data. Data can be accessed on request to the corresponding author after reports related to the primary outcomes are published.

### Patient and public involvement

Patients and/or the public were not involved in the design, or conduct, or reporting, or dissemination plans of this research.

### Strength and limitations

A primary outcome of older patients after cardiac surgery that receives special focus is morbidity or mortality; however, more recently, healthy life expectancy and quality-adjusted life years are also becoming important targets of treatment. Functional status or long-term care are, therefore, recognised as essential outcomes for promoting informed surgical decision for patients with cardiovascular diseases, their family and the surgeon. Despite extensive investigations into morbidity or mortality, very few studies have specifically investigated the risk of functional decline and long-term care dependency

following cardiovascular surgery. To our knowledge, the present study is the first large-scale, prospective, multicentre cohort study to examine the association of functional trajectory during hospitalisation and postdischarge regular exercise with functional decline and long-term care dependency. This study may suggest novel strategies to prevent functional decline and long-term care dependency in older patients after cardiac surgery.

The prognostic impact of HAD has recently been suggested in community-dwelling older adults or older patients with heart failure. However, the effect of HAD on long-term functional decline and long-term care dependency remains unknown. Moreover, another purpose of this cohort study is to examine the effects of regular exercise including CR on functional decline and long-term care dependency. In particular, exercise-based CR has favourable effects on functional status in older patients with poor functional status. Our large cohort study allows us to analyse the effects of outpatient CR on functional trajectory in patients with HAD.

In addition, physical as well as psychological and social aspects are important to maintain healthy life expectancy or quality-adjusted life years in older patients. Cognitive decline or decline in the social aspect may aid in the assessment of functional decline and long-term care dependency. Based on these findings, we assessed physical status, cognitive status and social status from KCL subcomponent scores.

This study is not population based due to the recruitment scheme in the collaborative hospitals that focuses on research and education in the field of cardiac surgery rehabilitation. Thus, this study is limited in its generalisability and underestimates the prevalence of functional decline. However, all facilities performed a large number of cardiac surgeries on older patients, normative older populations are also included in the study. Another limitation is selection bias introduced by the inclusion of individuals who meet the eligibility criteria and are capable of functional assessments or answering the questionnaires on functional status.

## ETHICS AND DISSEMINATION

The present study was organised according to the Guidelines for Epidemiological Research proposed by the Japanese Ministry of Health, Labour and Welfare. The study protocol was approved by the Ethics Committee of the Department of Physical Therapy (approval no. 20-030), Faculty of Health Science, Juntendo University, and of each collaborating hospital. We obtained written informed consent from all study participants after the description of the study procedures. The informed consent form states that all clinical information is confidential and will not be shared and that data are stored in a secure web-based database (REDCap). Patient recruitment began in April 2021, and the 2-year follow-up is expected to be completed in 2024. The study results will be published in a peer-reviewed journal, in addition to being presented at national and international conferences. We anticipate these results to be published in 2024, after completion of the 2-year follow-up of all recruited patients. Our findings will help inform future recommendations of postoperative care planning for older patients after cardiac surgery.

This study has an important potential limitation is selection bias introduced by the inclusion of individuals who meet the eligibility criteria and are theoretically capable of functional assessment or answering the questionnaire on functional status and regular exercise, as well as those with severe cognitive impairment. This may lead to the increased likelihood of other risk factors of functional decline and long-term care dependency and clinical events.

### Author affiliations
[1]Department of Physical Therapy, Faculty of Health Science, Juntendo University, Tokyo, Japan
[2]Department of Rehabilitation, Juntendo University Hospital, Tokyo, Japan
[3]Department of Rehabilitation, The Cardiovascular Institute, Tokyo, Japan
[4]Department of Rehabilitation, Saitama Medical University International Medical Center, Hidaka, Japan
[5]Department of Physical Therapy, Higashi Takarazuka Satoh Hospital, Takarazuka, Japan
[6]Department of Rehabilitation, Kobe City Medical Center General Hospital, Kobe, Japan
[7]Department of Rehabilitation, Fukuyama Cardiovascular Hospital, Fukuyama, Japan
[8]Department of Rehabilitation, Tsuchiya General Hospital, Hiroshima, Japan
[9]Department of Rehabilitation, Ryukyus University, Nishihara-cho, Japan

**Contributors** MS, TM and TT conceived the study, obtained grant funding and drafted the protocol. AM, AS, GT, HW, KI, KS, MT, YH and YO reviewed and revised the protocol.

**Funding** This work is supported by the MEXT KAKENHI Grant Number JP 20H04055.

**Competing interests** None declared.

**Patient and public involvement** Patients and/or the public were not involved in the design, or conduct, or reporting, or dissemination plans of this research.

**Patient consent for publication** Consent obtained directly from patient(s).

**Provenance and peer review** Not commissioned; externally peer reviewed.

**ORCID iD**
Masakazu Saitoh http://orcid.org/0000-0001-8666-6353

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
