## [Reviewer comments · BMJ Open]

ARTICLE DETAILS

TITLE (PROVISIONAL)	Protocol for a multicentre, prospective observational cohort study in Japan: association among hospital-acquired disability, regular exercise, and long-term care dependency in older patients after cardiac surgery
AUTHORS	Saitoh, Masakazu; Takahashi, Tetsuya; Morisawa, Tomoyuki; Sakuyama, Akihiro; Watanabe, Hidetaka; Sakurada, Koji; Hanafusa, Yusuke; Tahara, Masayuki; Iwata, Kentaro; Ochi, Yusuke; Takamura, Go; Minei, Akira

VERSION 1 – REVIEW

REVIEWER	Harky, Amer Liverpool Heart and Chest Hospital NHS Foundation Trust
REVIEW RETURNED	19-Jul-2021

GENERAL COMMENTS	A very important study as the population for cardiac surgery grows and mostly are fragile patients. My question is: why 2 years of follow-up and not 5 years? furthermore, do you have any clear cut of frailty score for such patients? such as STS scores? Finally, are you considering only elective CABG patients? any age limit?
---

REVIEWER	Olsson, Anders Karolinska Institute, Department of Clinical Science and Education
REVIEW RETURNED	31-Oct-2021

GENERAL COMMENTS	Thanks for the opportunity to review this study protocol. This study aims at answering an important question. However I have two queries. 1. How common is cardiac surgery in Japan? Is the study sample representative for the normative elderly population. This may be discussed in "background".2. There are several evaluation methods used in this study with some possible overlapping of methods. This may affect the capacity for the participants to fulfill the examinations. This may be discussed.
---

VERSION 1 – AUTHOR RESPONSE

Response to Reviewer: 1

Thank you for reviewing and constructive comments to improve the quality of our manuscript.

Comments 1

My question is: why 2 years of follow-up and not 5 years? furthermore, do you have any clear cut of frailty score for such patients? such as STS scores?

Answer to Comment 1

Thank you for your comment.

Several reports have been demonstrated that the long-term prognosis is good even in older patients after cardiac surgery in Japan. However, in recent years, a high prevalence of hospital-acquired disability or poor functional recovery after discharge has been shown in the older population^{1,2}. We also assume that the progression of long-term care dependency begins gradually after hospital discharge. As pointed out by the reviewer¹, long-term follow-up is important, however, the aim of the study was early detection of function decline or progression of long-term care dependency, and to investigate the preventive effect of regular exercise. Therefore, we have set a 2-year follow-up period for this study.

We used the Kihon checklist (KCL), a frailty score that has been validated through studies such as original Fried criteria^{3,4}. KCL score of 8 or more points indicates a diagnosis of frailty.

- 1) Loyd C, et al. Prevalence of Hospital-Associated Disability in Older Adults: A Meta-analysis. *J Am Med Dir Assoc.* 2020; 21(4): 455-461.e5.
- 2) Freedman VA, et al. Recent trends in disability and functioning among older adults in the United States: A systematic review. *JAMA* 2002; 288: 3137.
- 3) Satake S, et al. Validity of the Kihon Checklist for assessing frailty status. *Geriatr Gerontol Int.* 2016; 16(6): 709-15.
- 4) Sewo Sampaio PY, et al. Systematic review of the Kihon Checklist: Is it a reliable assessment of frailty? *Geriatr Gerontol Int.* 2016; 16(8): 893-902.

Comment 2

Finally, are you considering only elective CABG patients? any age limit?

Thank you for your comment.

Answer to Comment 2

The study is a multicentre, prospective cohort study enrolling older patients aged ≥ 65 years and scheduled to undergo elective cardiac surgery.

This information is described in the Participant eligibility criteria section as follows:

(Page 5, L16-20)

Participant eligibility criteria

Potential participants are recruited from collaborating hospitals. To be eligible for study participation, patients must be aged ≥ 65 years and scheduled to undergo elective cardiac surgery. Exclusion criteria are presented in Figure 2. All patients provided informed consent at the time of recruitment.

Answer to Reviewer: 2

Thank you for reviewing and constructive comments to improve the quality of our manuscript.

Comment 1

How common is cardiac surgery in Japan? Is the study sample representative for the normative elderly population. This may be discussed in "background".

Answer to Comment 1

Thank you for your helpful comment.

Cardiac surgery is the most common thoracic surgery in Japan. The number of cardiac surgeries for older patients is increasing, and challenges are functional decline or progression of long-term care dependency after hospital discharge.

On the other hand, the study sample in our study limits its generalizability. Therefore, we revised the Introduction section and Strength and Limitation section according to your comments as follows:

(Abstract: Page2, Line 3-6)

Cardiac surgery for older patients, postoperative functional decline and the need for long-term care have received increasing attention as essential outcomes in recent years.

(Introduction: Page 4, L 4-8)

The Japanese Association for Thoracic Surgery has conducted annual surveys of thoracic surgery throughout Japan. Cardiac surgery is most common of thoracic surgery, more than 70,000 patients in Japan undergo cardiac surgery annually [3], in particularly the absolute number of older patients undergoing cardiac surgery is increasing [4].

(References: Page11, Line 29-31)

We added a new reference to Reference List as follows:

3. Shimizu H, Okada M, Tangoku A, et al. Thoracic and cardiovascular surgeries in Japan during 2017 : Annual report by the Japanese Association for Thoracic Surgery. *Gen Thorac Cardiovasc Surg.* 2020;68(4):414-449.

(Strength and limitations of this study: Page 3, L9-12)

This study is not population based due to the recruitment scheme in the collaborative hospital that focuses on research and education in the field of cardiac surgery rehabilitation, which limits its generalisability and underestimate the prevalence of functional decline.

Comment 2

There are several evaluation methods used in this study with some possible overlapping of methods. This may affect the capacity for the participants to fulfill the examinations. This may be discussed.

Answer to Comment 2

Thank you for your helpful comments.

As reviewer 2 pointed out, we used several functional assessments.

BI was used as an indicator of basic ADLs, and to identify Hospital-acquired disability (HAD). J-CHS is a physical frailty assessment. On the other hand, KCL is a comprehensive frailty assessment that includes physical, mental, and social frailty. KCL and J-CHS are different measures of frailty. SPPB is a mobility assessment and grip strength is a muscle strength assessment. These assessments are measurable in all patients except those with cognitive impairment. However, this may affect the capacity of the participants to fulfil the examinations. We therefore revised the Strength and Limitation section as follows:

(Strength and limitations of this study: Page 3, L 13-15)

- An important potential limitation is selection bias introduced by the inclusion of individuals who meet the eligibility criteria and are capable of functional assessment or answering the questionnaire on functional status.

(Strength and Limitations: Page 10, L 11-18)

This study is not population based due to the recruitment scheme in collaborative hospitals that focuses on research and education in the field of cardiac surgery rehabilitation. Thus, this study is limited in its generalisability and underestimate the prevalence of functional decline. However, all facilities performed a large number of cardiac surgeries on older patients, normative older populations are also included in the study. Another limitation is selection bias introduced by the inclusion of individuals who meet the eligibility criteria and are capable of functional assessments or answering the questionnaires on functional status.